# Dysregulation of the Arachidonic Acid Pathway in Cystic Fibrosis: Implications for Chronic Inflammation and Disease Progression

**DOI:** 10.3390/ph17091185

**Published:** 2024-09-09

**Authors:** Simona D’Orazio, Domenico Mattoscio

**Affiliations:** 1Department of Medical, Oral and Biotechnology Sciences, University “G. d’Annunzio” Chieti-Pescara, 66100 Chieti, Italy; simona.dorazio@unich.it; 2Center for Advanced Studies and Technology (CAST), University “G. d’Annunzio” Chieti-Pescara, 66100 Chieti, Italy

**Keywords:** arachidonic acid, cystic fibrosis, CFTR, DHA, lipid peroxidation, prostaglandins, leukotrienes, cyclooxygenases, lipoxygenases, eicosanoids

## Abstract

Cystic fibrosis (CF) is the most common fatal genetic disease among Caucasian people, with over 2000 mutations in the CFTR gene. Although highly effective modulators have been developed to rescue the mutant CFTR protein, unresolved inflammation and persistent infections still threaten the lives of patients. While the central role of arachidonic acid (AA) and its metabolites in the inflammatory response is widely recognized, less is known about their impact on immunomodulation and metabolic implications in CF. To this end, here we provided a comprehensive analysis of the AA metabolism in CF. In this context, CFTR dysfunction appeared to complexly disrupt normal lipid processing, worsening the chronic airway inflammation, and compromising the immune responses to bacterial infections. As such, potential strategies targeting AA and its inflammatory mediators are being investigated as a promising approach to balance the inflammatory response while mitigating disease progression. Thus, a deeper understanding of the AA pathway dysfunction in CF may open innovative avenues for designing more effective therapeutic interventions.

## 1. Introduction

Cystic fibrosis (CF) is the most common fatal genetic disease in Caucasians, with an incidence of 1 in 3200 births in the North European population. The basic genetic defect has been localized in the cystic fibrosis transmembrane conductance regulator (CFTR) gene, with more than 2000 mutations identified so far [1]. The CFTR gene product is primarily a cAMP-activated chloride ion channel, which is highly expressed by epithelial cells but also on endothelial cells [2], polymorphonuclear neutrophils (PMN) [3], monocytes/macrophages [4], dendritic cells [5], platelets [6], and lymphocytes [7]. CFTR mutations alter the fluid and electrolyte composition of secretions and the homeostasis of organs such as lung, pancreas, and liver, and sustain an unrelenting inflammatory reaction. Although CF affects almost every single tissue and organ in the body, chronic airway infection and inflammation, leading to pulmonary insufficiency, is responsible for at least 80% of CF-related deaths. However, disease severity is highly dependent on genetic variability and environmental factors (recently reviewed in [8]).

Under physiological conditions, acute inflammation requires timely resolution to ensure the return to tissue homeostasis after the removal of the biological threat. During resolution, actively regulated by several mechanisms, including the production of specialized pro-resolving lipid mediators (SPM), pro-inflammatory mediators are catabolized, PMN recruitment decreases, and return to tissue homeostasis is promoted by macrophages [9]. Failure of effective resolution leading to persistent inflammation, tissue damage, and organ fibrosis has been documented also in CF (recently reviewed in [10,11,12]). 

Although progress in multidisciplinary care and the introduction of modulators [13] have led to a substantial increase in the median life expectancy of people with CF, the current strategies may not completely restore several molecular and clinical defects [14,15]. Therefore, new pharmacological approaches to manage inflammation in CF are still required. 

In addition to cytokines, chemoattractants, and SPM, oxylipins are key regulatory players in the inflammatory response and its resolution. Oxylipins are oxygenated products of polyunsaturated fatty acids (PUFAs), which derive from the catabolism of the essential PUFA linoleic acid (LA) and alpha-linolenic acid (ALA) [16]. Arachidonic acid (AA) is widely recognized as the key oxylipin precursor driving the inflammatory process [17]. Thus, an imbalance in AA metabolism may sustain CF inflammation. Indeed, alterations in AA precursors and metabolites have been documented in CF [18]. Therefore, a comprehensive understanding of the dysfunctional AA pathway in CF is relevant to gain further knowledge about CF lung inflammation. 

This review aims to summarize how dysfunctions of AA metabolism, related to CFTR-loss-of-function, fuel CF lung disease. Furthermore, novel AA-based approaches to prevent the progression of CF disease will be discussed.

## 2. Alteration of the AA Pathway in CF, from Precursors to Metabolites

AA is an omega-6 fatty acid (20:4 n-6 with 20 carbon atoms and 4 double bonds, the last of which is in position 6 (omega) from the terminal carboxyl group) which is key in the inflammatory response. In the following sections, we will describe the mechanisms of altered PUFA, AA, its metabolites, and enzymes, and how they contribute to CF airway pathology.

### 2.1. n-3 and n-6 PUFA

AA is derived from linoleic acid (LA), the essential fatty acid, along with alpha-linolenic acid (ALA). Although humans cannot synthesize essential fatty acids but absorb them with the diet, the organism has developed a series of enzymes with desaturating and elongating activity capable of converting LA to the omega-6 AA, and ALA to omega-3 eicosapentaenoic acid (EPA), which is further metabolized to docosahexaenoic acid (DHA) (Figure 1). Once synthesized, AA, EPA, and DHA become membrane phospholipids, the concentration of which depends on nutritional status, metabolic processes, specific cell/tissue type, enzyme activity, and disease state [19,20].

Since the 1970s [21], alterations in fatty acid metabolism have been widely reported in CF and recognized early in infants with CF and animal models of CF, indicating deficiencies owing to the genetic defect rather than secondary to pancreatic insufficiency or malnutrition [22,23]. Among fatty acids, cord-blood samples from CF newborns showed the absence or reduced levels of gamma-LA (a metabolic product of LA), alfa-LA (a precursor of DHA), and DHA, with an overall reduction in n-3 metabolites as compared to healthy newborns [22]. Similarly, the plasma of adults with CF showed LA and DHA deficiency [24], specifically due to defective essential fatty acid metabolism [24,25,26], consistently reproduced in mouse models (reviewed in [27]). Importantly, marked LA deficiency was associated with more severe CFTR mutations and phenotypes, indicating a crucial impact of this alteration on disease progression [27,28]. As discussed below, in addition to strikingly affecting the CF inflammatory response, this PUFA unbalance impaired plasma membrane fluidity [26], with pathobiological consequences that need to be elucidated. Importantly, n-3 and n-6 PUFA form highly specific interactions that affect their distribution in the cell membrane, and thus their biological effects depend on their cellular ratio rather than on their absolute levels [29]. Studies conducted on CFTR-expressing tissues in people with CF and rodents identified an elevated AA/DHA ratio in CF samples [30,31,32]. Corroborating the strict interdependence and the fine balance between bioactive lipids, DHA supplementation induced both a significant increase in DHA and a decrease in AA levels in the lungs and pancreas of treated mice, as well as in CF in vitro systems and in patients [32,33,34]. Thus, LA and ALA may share an unbalanced metabolism followed by dysregulated levels of both AA and DHA in CF. Indeed, most of the enzymes involved in the AA/EPA/DHA pathways are mutual and compete for metabolism (Figure 1). In line with this, a large body of evidence points to alteration in desaturase enzymes as responsible for the reduced LA and DHA levels found in CF [35]. Moreover, reduced LA correlated with boosted expression and activity of the Δ5- and ∆6-desaturase [35], suggesting that alteration in LA metabolism in CF is due to overreactive desaturases driven by the CFTR defect [36]. Supplementation with DHA reversed desaturase hyperactivity and restored intrinsic alteration of the PUFA unbalance in CF in vitro [33].

However, even if it is clear that DHA is reduced in CF, mechanisms leading to such reduction are less well understood. Since Δ6-desaturase activity is shared, it is surprising that LA levels are decreased in CF tissues, but DHA levels, the product of EPA metabolized by the same enzyme, are not. A mechanistic explanation of this apparent paradox was obtained in CF epithelial cell lines, where the conversion rate of EPA to DHA is reduced while the conversion of LA to AA is increased [35]. On the other hand, recent studies have reported that different isoforms of Δ6-desaturase and Δ5-desaturase may exist, so AA and EPA/DHA may be differentially metabolized within the cell [37]. 

### 2.2. Arachidonic Acid

As mentioned above, AA concentration is higher in CF tissues due to the enhanced metabolism of LA to AA. This increase is independent of persistent *P. aeruginosa* infection or chronic inflammation but is rather CF-specific [38], suggesting a CFTR-dependent alteration. Indeed, increased phospholipid-bound AA levels and decreased DHA levels were only found in the lung, pancreas, and ileum of CFTR −/− mice compared to healthy controls [18]. Similarly, the AA/DHA ratio was elevated in canonical CFTR-expressing tissues such as mucosal and submucosal nasal and rectal specimens in patients with CF [30]. Mechanistically, in addition to the increased conversion of LA to AA [35], the increase in free AA could also be due to enhanced release from the plasma membrane by the activity of the membrane-releasing enzyme cytosolic phospholipase A2 (cPLA2) [39,40]. Hyperactivation of cPLA2 may contribute to the establishment of the inflammatory environment in the CF airways even in the absence of infection. Indeed, mutant CFTR loses its inhibitory activity against cPLA2 by disrupting the Annexin A1/CFTR/cPLA2 complex at the plasma membrane. The hyperactivated cPLA2 drives the excessive AA release from the plasma membranes of CF cells [41]. In addition to its role in regulating cPLA2 activation, annexin-1 is an anti-inflammatory protein downregulated in the nasal epithelial cells of patients with CF and the lung and pancreas of CFTR KO mice [42]. Therefore, reduced annexin-1 expression driven by the basic CFTR genetic defect sustains chronic inflammation [43] and disrupts the cPLA2-CFTR assembly at the plasma membrane, finally triggering cPLA2 hyperactivation and AA release.

In addition to altering eicosanoid levels (see the following paragraph), the increased AA concentration modifies plasma membrane composition and fluidity, the mobility of membrane-bound proteins, and ion channel activity [44]. Notably, AA, applied to the cytoplasmic side of excised membrane patches of baby hamster kidney cell lines, inhibited CFTR-dependent chloride flux by electrostatic interaction with cytoplasmic amino acid side chains of the CFTR channel pore [45,46]. Thus, the aberrant AA concentration in the CF airway may further contribute to aggravating the CFTR channel dysfunction. Along these lines, lipidomic analysis of CF bronchial epithelial cell lines showed lipid modifications in the cell membrane due to the altered balance between phospholipid types [47]. Compared to the wild type, CF cells showed a higher accumulation of ceramide and glycosylated sphingolipids, which are known to be involved in inflammatory processes [48]. Hence, these findings in CF models suggest that alterations in the AA compartment may impair both chloride efflux and plasma membrane dynamics, adding further levels of complexity to the understanding of the role of AA in CF. 

### 2.3. Metabolites

Among PUFAs, the biological functions of AA are closely related to inflammation since it is the substrate of both a class of potent inflammatory lipids, the eicosanoids, and lipoxins, a family of SPM [49]. Thus, a balanced AA metabolism is crucial to maintain a homeostatic regulation of the inflammatory response. For this reason, the biological synthesis and degradation of AA-derived oxylipins must be tightly regulated. AA metabolism initiates with the release of membrane-bound AA into the cytoplasm by the action of cPLA2, which catalyzes the hydrolysis of the acyl bond between the second fatty acid tail and the glycerol molecule of membrane phospholipids. cPLA2 is activated by inflammatory stimuli and its activity is the rate-limiting step in the AA pathway [50]. Once free, AA is the substrate of three families of oxylipin-generating enzymes: cyclooxygenases (COX), responsible for the synthesis of prostanoids; lipoxygenases (LOX) that catalyze the biosynthesis of leukotrienes and lipoxins; and CYP450 epoxygenases form the epoxyeicosatrienoic acids (EET). In addition, AA can be non-enzymatically oxidized by reactive oxygen species to isoprostanes (recently reviewed in [19]) (Figure 1). 

Lung inflammation and parenchymal injury are early detectable in CF, with accumulation of membrane bioactive lipids and AA metabolites. Indeed, in children with CF, eicosanoid levels (prostaglandin G_2_ and isoprostanes) in bronchoalveolar lavage fluids (BAL) were high and correlated with markers of airway inflammation such as PMN elastase and myeloperoxidase, and with lung damage [51]. Therefore, the alteration of AA metabolism deeply impacts CF inflammation and outcomes. In the following sections, we will examine the main pathways involved in AA-derived oxylipin production, presenting the most relevant changes occurring in CF. 

#### 2.3.1. The COX Pathway

AA can be transformed by COX-1 or COX-2 into prostaglandins (PG) or thromboxanes (Tx) (Figure 1). Both COX isoforms release an oxygenated partially cyclic lipid, However, COX-1 is constitutively expressed while COX-2 is inducible by inflammatory stimuli, hormones, and growth factors [19].

One of the key COX products is PGE_2_, a central bioactive lipid at the interface between the pro-inflammatory and pro-resolving response. PGE_2_ is released by immune cells during the onset of the acute inflammatory response, as it regulates vascular permeability, chemotaxis, and pro-inflammatory cytokine release. Secondly, PGE_2_ is also required to boost the release of lipoxins (LX), bioactive AA metabolites known for their pro-resolving and anti-inflammatory functions [52]. PGD_2_ and TxA_2_ could be also produced from the same PGE_2_ precursor (PGH_2_). PGD_2_ is the major COX-metabolite produced by activated macrophages and mast cells, while TxA_2_ is mainly released by activated platelets, monocytes, macrophages, PMN, and lung parenchyma during inflammation [19].

Marked staining for COX-2 has been found in the sinonasal tissue of people with CF, underlining its upregulation in the upper airways due to post-transcriptional mechanisms [53,54], also confirmed by in vitro assays on cell cultures [35]. COX-1 overexpression was also found in nasal CF polyps [54], even if its overexpression in other CF tissues is controversial [53]. As a consequence, patients with CF had excessive levels of COX-1 products such as PGE_2_ in saliva, urine, sputum, and BAL and PGD_2_ in sputum [55,56,57,58], suggesting their potential role in driving airway inflammation. Notably, PGE_2_ but not PGD_2_ levels correlated with the CF genotype and disease severity. Higher levels of urinary PGE_2_ and PGD_2_ metabolites were detected in patients with class I, II, and III mutations and with bronchiectasis as compared to control healthy volunteers [59,60]. Consistent with this, a longitudinal study on a cohort of children with CF showed that PGs correlated with structural lung damage and predicted the evolution of lung disease within two years [51]. 

TxA_2_ is also a key COX-derived product involved in CF lung chronic inflammation. Indeed, the concentration of its stable metabolite, TxB_2_, was consistently found to increase in the urines of patients with CF and correlated with pulmonary dysfunction [56,61]. As TxB_2_ excretion is considered an in vivo index of platelet activation [62], high TxB_2_ concentrations were correlated to sustained platelet activation that could be partially corrected by antioxidant supplementation [61]. 

#### 2.3.2. The LOX Pathway

The involvement of AA metabolism in both pro-inflammatory and pro-resolving responses is well illustrated by the LOX pathway. There are several isoforms of this enzyme, namely 5-, 12-, and 15-LOX. Before being converted to leukotrienes or lipoxins, AA is oxidized to chiral-specific unstable intermediates denominated hydroperoxyl eicosatetraenoic acids: 5(S)-HpETE from 5-LOX, 12(S)-HpETE from 12-LOX, and 15(S)-HpETE from 15-LOX. The reduced products of HpETEs are hydroxy eicosatetraenoic acids (HETEs), which are secreted by epithelial cells and leukocytes during the inflammatory response (Figure 1). Overall, HETEs are associated with pro-inflammatory events, e.g., 5(S)-HETE and 15-(S)HETE are potent chemoattractants for PMN and stimulate their degranulation [63]. On the other hand, 15(S)-HETE inhibited PMN transepithelial migration, degranulation, and superoxide production [64]. Alterations in HETEs have been described in several chronic inflammatory diseases such as obesity, cardiovascular disease, and cancer [65], but little is known about HETE alterations in CF. Since 15(S)-HETE is the predominant LOX metabolite in the human lung [66], it may play a critical role in the regulation of the inflammatory response in CF. Thus, further investigations are needed in this field.

With 5-HpETE as an intermediate, 5-LOX converts AA to leukotriene A_4_ (LTA_4_). This bioactive lipid mediator is best known as a sparkplug of acute inflammation, acting as a chemoattractant for innate immune cells, but also for Th17 lymphocytes [67]. Apart from its physiological role, LTA_4_ could perpetuate the inflammatory state, leading to chronicity and further tissue damage in diseases such as asthma and atherosclerosis [68]. LTA_4_ is the precursor of two major classes of leukotrienes, cysteinyl LTs and LTB_4_. Specifically, LTA_4_ is converted by LTC_4_ synthase to LTC_4_ and finally to the stable metabolites LTD4 and LTE_4_. These metabolites are collectively termed cysteinyl leukotrienes (CysLT) (Figure 1). CysLT are contractile agonists of airway smooth muscles and stimulate mucus production by goblet cells, resulting in bronchoconstriction and mucus accumulation, key events in CF lung disease [69]. On the other hand, LTA_4_ hydrolase converts LTA_4_ into LTB_4_, a potent chemoattractant for leukocytes during immune responses (reviewed in [70]).

LT in CF airways significantly contribute to airway inflammation. Indeed, LTB_4_ and CysLT in the sputum of children with CF correlated with tumor necrosis factor-alpha (TNF-α) levels and parameters of airflow obstruction [71], pointing to key roles for these mediators in sustaining CF inflammation. Indeed, analysis of CF sputum and epithelial lining fluid revealed LTB_4_ as a major AA metabolite along with CysLT [57,72]. Elevated levels of LTB_4_ and CysLT are early detectable in the disease, as they were found at nanomolar concentrations in the sputum and urine of CF infants [73]. This evidence supports the hypothesis that the inflammatory response could be driven by altered lipid metabolism even in the absence of infection. 

Actions of 12-LOX and 15-LOX, in cooperation with 5-LOX, can convert AA into lipoxins (LXs). Two main routes have been described for the generation of LXA_4_ and its positional isomer LXB_4_: the 5/15-LOX and the 5/12-LOX pathways (reviewed in [74]). The 5/15-LOX pathway can be either initiated by 5-LOX or 15-LOX in PMN, eosinophil or alveolar macrophages, to convert AA into intermediates that are transformed by the reciprocal LOX into both LXA_4_ and LXB_4_. Transcellular interactions between PMN and eosinophils or PMN and lung cells has also been described to generate LX through 5/15-LOX metabolism. Conversely, 5-LOX from PMN generates and transfers LTA_4_ as a precursor for platelets, which convert LTA_4_ into LXA_4_ and LXB_4_ through 12-LOX activity (Figure 1) [75,76]. The biological significance of this transcellular interaction is extremely important, as the physical proximity of inflammatory and resident cells at the site of injury promotes the initiation of specific signals toward resolution and homeostasis. Finally, in a third biosynthetic pathway involving transcellular interactions among PMN and endothelial or tumor cells [77,78], as well as hepatocytes and liver cells [79], COX-2 acetylated by aspirin produces 15(R)-HETE, which is converted by 5-LOX in 15-epimers of LXA_4_ and LXB_4_ (named aspirin-triggered lipoxins, ATL). As LX, ATL generate a range of anti-inflammatory actions, i.e., reduction in PMN migration and activation, and pro-resolving responses by supporting efferocytosis and active removal of pro-inflammatory mediators and debris [37,76,80,81].

LXA_4_ levels are significantly low in patients with CF, both adults and children. As compared to patients with other inflammatory lung diseases, people with CF had the lowest LXA_4_ concentration in airway fluids that contribute to maintain airway inflammation. Indeed, administration of a stable LXA_4_ analog to a chronically infected mouse model of CF ameliorated markers of inflammation and lung functions [82]. Mechanistically, this failure in LXA_4_ production could be ascribed to the downregulation of 15-LOX expression observed both in CF epithelial cells [83] and BAL from children with CF even in the absence of detectable infection [84]. On the other hand, CFTR loss-of-function deeply affects the activity of platelet 12-LOX, reducing considerably LXA_4_ and LXB_4_ generation during PMN/platelet coincubations [6]. In addition to the reduced LX production, CF macrophages showed lower expression of LXA_4_ receptor FPR2/ALX compared to cells from healthy volunteers. Importantly, restoration of normal levels of FPR2/ALX improved phagocytosis by CF macrophages, indicating the crucial role of LXA_4_ signaling in mechanism of CF inflammation [85]. Therefore, defects in LOX expression, activity, metabolites, and their receptors greatly contribute to impair resolution of inflammation in CF lung.

#### 2.3.3. The CYP450 Pathway and Oxidative Stress

While the COX and LOX pathways are quite well studied in CF, less is known about the AA metabolites formed by CYP450. Once recognized by this class of enzymes, AA and its intermediates are epoxidized to EET, high-affinity ligands of PPAR-γ [86], and are generally involved in the anti-inflammatory response [87] (Figure 1). Thus, investigating the extent to which CYP450 and EETs are involved in non-resolving inflammation in CF may provide further information on lipid imbalance and potential therapeutic approaches.

Interestingly, a recent study reported an AA-oxylipin produced by *P. aeruginosa* [88], one of the most prevalent colonizing bacteria of CF airways [89]. Indeed, *P. aeruginosa* contains CYP168A1, a cytochrome P450 enzyme with high affinity for AA, which is readily found at high concentrations in the CF host environment. The main product of CYP168A1 catalysis is 19-HETE, a potent vasodilator and inhibitor of platelet aggregation [90], that facilitates blood vessel permeability and leukocyte transmigration. The reason beyond the release of 19-HETE by *P. aeruginosa* remains to be elucidated, but its ability to modulate the immune response by acting on host lipid mediator signaling may be fundamental for its persistent colonization, which remains a life-threatening condition in the compromised airways of people with CF. 

The impact of oxidative stress on CF pathophysiology is receiving increasing attention. Accumulating evidence shows how the extent of pathological markers in CF lung parallels that of oxidative markers, during both chronic and acute exacerbations and disease progression [91]. PMNs are recognized as a major contributor to the exaggerated release of reactive oxygen species (ROS) during infections, due to failure of phagocytosis and continuous contact with bacteria in CF [92,93]. Furthermore, epithelial cells support the oxidative burst by continuously releasing ROS [94], even in the absence of infection. Worsening this scenario, patients with CF also showed impaired antioxidant detoxification mechanisms, such as extracellular glutathione (GSH) deficiency [95]. A contribution to lipid peroxidation also comes from pancreatic insufficiency, which could be responsible for the malabsorption of essential fat-soluble antioxidants [91]. Despite the generating mechanism, one immediate consequence of altered oxidant/antioxidant ratio is the ROS reaction with cellular membranes and lipoproteins [96]. Polyunsaturated fatty acid phospholipids are highly susceptible to oxidation with conversion to bioactive or toxic lipids called isoprostanes [97]. Because of their origin, they are considered markers of oxidative stress also in lung diseases [98]. In physiological conditions, isoprostanes are biologically active and are potent mediators of inflammation [99]. For instance, they can bind to TLR2 on macrophages and trigger the expression of COX-2 and interleukin 1-β, promoting inflammation, but they could also boost pathogen clearance and removal of apoptotic cells [100].

The presence of isoprostanes in the lungs of patients with CF can be detected at an early age, as found in BAL fluid from children aged 3–5 years [51]. The gold standard used as a reliable measure of lipid peroxidation is the isoprostane 8-iso-PGF-2α, whose levels positively correlate with markers of lung inflammation, disease severity [101], and exacerbations [102,103]. Remarkably, antibiotic therapy against *P. aeruginosa* infection does not alter levels of 8-iso-PGF2α, cys-LTs, and PGE_2_ in the airways, suggesting that, despite the decreased bacterial load and improved clinical status, there is persistent lipid peroxidation of AA [102,104]. In addition to lungs, elevated isoprostane levels have been found in the urine and plasma of people with CF compared to healthy controls [61,105], underlying a systemic circulation of oxidized lipids that could further affect peripheral tissues. 

Moreover, also the LOX products HETEs can be non-enzymatically oxidized to form keto-phospholipids (KETEs) [106]. In a cohort including healthy subjects, people with CF, and people with other lung diseases, such as bronchiectasis and persistent bacterial bronchitis, CF samples showed significantly increased 15-KETE levels [106]. However, their role remains to be fully elucidated.

Figure 2 summarizes the alterations in AA precursors, metabolites, and enzymes found in CF, along with the reported molecular mechanisms leading to dysfunction.

### 2.4. Role of CFTR Dysfunction in CF Abnormal Fatty Acid Metabolism

As previously highlighted, CFTR loss-of-function directly impacts AA production and metabolism (Figure 3). However, with the introduction of highly effective modulator therapy to rescue CFTR functions, it becomes crucial to determine whether this therapy modulates the unbalanced AA pathway. Although CFTR could control AA metabolism, bacterial antigens, pro-inflammatory cytokines, and dysfunctional immune cells may also significantly contribute to increase eicosanoid levels in the inflamed CF lung [107]. 

As mentioned, the deficiency of plasma LA and DHA, together with the excessive release of AA, has been positively correlated with genotype severity in patients with CF [108]. The altered circulating AA/DHA ratio may reflect the alterations in membrane phospholipids [24]. CFTR is also localized at the cell membrane and, as a channel, can interact with phospholipids and affect their trafficking, incorporation, and turnover [109]. Thus, CFTR may contribute to the regulation of DHA and AA processing [32]. Indeed, in newborn pigs, CFTR loss caused blunted lung uptake of fatty acids, even if their circulating levels were high. Interestingly, AA was the only PUFA with an increased uptake, suggesting a specific mechanism driven by CFTR that directs AA from the bloodstream to the lung [110]. Consistent with this, the increased AA/DHA ratio occurs early in life due to changes in lipid membrane dynamics caused by CFTR loss-of-function. In line with the role of CFTR in regulating membrane composition, in vitro and in vivo CFTR knockout models highlighted the impact of CFTR on the organization of lipid rafts in the cell membrane, affecting both phospholipid and protein distribution. In human bronchial epithelial cells, CFTR loss-of-function increased the conversion of LA to AA and decreased DHA generation from precursors [36]. Similarly, FF508del CFTR enhanced the conversion of LA to AA compared to cells with wild-type CFTR [111]. Therefore, CFTR mutations directly affect AA metabolism by modulating membrane lipid dynamics. In line with this, CFTR inhibition caused membrane destabilization with disrupted formation of protein complexes, increased cPLA_2_ activity, and eicosanoid biosynthesis in airway epithelial cells [41], suggesting a direct relationship between CFTR, plasma membrane, and AA metabolism.

In addition to membrane dynamics, CFTR could also directly modify, mainly at the mRNA level, the expression of enzymes involved in AA metabolism, such as desaturases and COX-2 [54]. In addition to AA enzymes, a variety of changes in gene expression have been observed in cells with CFTR deletion [112], such as alterations in transcription factors, cytokines, and membrane receptors that are part of the inflammatory pathway. This reprogramming not only compensates for the absence of CFTR function but also suggests a direct role for CFTR as a transcriptional regulator perhaps through changes in protein–protein interactions and altered ion fluxes within the cell.

Importantly, CFTR dysfunction also causes the drastic reduction in antioxidant agents that act as scavengers of ROS, promoting lipid peroxidation and dysregulation of AA metabolism. Indeed, due to the impaired chloride efflux from CF cells, GSH transport in the extracellular environment is dampened (recently reviewed in [113]), thus contributing to enhance oxidative stress in CF. 

### 2.5. Therapeutic Strategies to Modulate AA Metabolism in CF

The advent of CFTR modulators has revolutionized CF care. These medications enhance CFTR protein expression and activity, leading to significant improvements in lung function, pulmonary exacerbations, and overall quality of life [114,115]. Ivacaftor, lumacaftor, tezacaftor, and elexacaftor are among the most widely used modulators, However, knowledge on their impacts on CF chronic inflammation, as well as their influence on AA metabolism, remains incomplete. 

Lipidomic characterization of CF cell lines treated with elexacaftor/tezacaftor/ivacaftor demonstrated that modulators could alter membrane and intracellular lipidome [116]. Indeed, an analysis from the GOAL study evidenced significantly reduced AA and PGE_2_ levels but not the rescue of LA and DHA deficiency in individuals with CF under ivacaftor [117], suggesting that CFTR correction could partially mitigate some aspects of the aberrant AA-derived inflammatory responses. Mechanistic studies reported that CFTR modulators lower AA levels and increase DHA levels in CF cell lines, but not in WT cells, thus confirming a direct impact of CFTR on lipid metabolism [118]. 

Dietary supplementation with omega-3 fatty acids, such as EPA and DHA, has been proposed as an encouraging therapeutic approach to reduce inflammation and balance the dysregulated AA/DHA ratio. These fatty acids can compete with AA for incorporation into cell membranes and enzymatic conversion, resulting in a shift toward the production of anti-inflammatory and pro-resolutive lipid mediators. Indeed, both adults and children with CF showed restoration of the lipid profile of red blood cell membranes, with reduced AA/EPA and AA/DHA ratios [119,120]. In line with this, DHA supplementation resulted in increased levels of DHA and its anti-inflammatory metabolites, such as 17-hydroxy-docosahexaenoic acid (17OH-DHA), along with a reduction in LTB_4_, 15-HETE, and PGE_2_ levels in the airways of adults with CF [119], confirming that DHA modifies lipid metabolism. However, DHA supplementation did not improve pulmonary function parameters such as forced vital capacity (FVC) and forced expiratory volume in one second (FEV_1_), likely because of the short duration of the study. Supporting this hypothesis, inflammatory mediators and DHA-derived metabolites returned to baseline after washout, suggesting the need for continuous DHA supplementation to maintain anti-inflammatory effects. However, even if long-term DHA (48 weeks) in patients with CF altered the fatty acid balance by increasing omega-3 levels and decreasing omega-6 levels, including AA, pulmonary and inflammatory markers (interleukin release, FEV_1_, or pulmonary exacerbations) were not affected [121]. Similarly, a meta-analysis of five randomized controlled trials with 106 participants evidenced low benefits for DHA supplementation in CF also because of the low quality of evidence across outcomes [122]. A recent randomized, double-blind, placebo-controlled study reported significant improvements in FVC and FEV_1_ in children with CF taking DHA [123]. In contrast, no significant differences were found in sputum and serum levels of inflammatory biomarkers, such as PMN elastase, resolvin D1, IL-8, and TNF-α, suggesting that the anti-inflammatory effects of DHA may be limited. Overall, additional studies are needed to understand optimal dosage, timing, need for concomitant administration of pancreatic enzymes, and categories of patients with patients with CF that may benefit from DHA supplementation.

Notably, since LA levels are decreased in patients with CF, LA supplementation has been proposed to correct this defect. However, evidence suggests that this supplementation may even exacerbate inflammatory responses. In fact, increased LA intake leads to higher levels of AA and its pro-inflammatory metabolites, exacerbating lung inflammation and CF lung disease [124]. 

As part of anti-inflammatory treatments to reduce abnormal fatty acid metabolism in CF, drugs that target specific enzymes involved in AA metabolism have been also used. For example, COX inhibitors such as non-steroidal anti-inflammatory drugs (NSAIDs) can reduce the synthesis of inflammatory PG. Among them, ibuprofen (COX-1 and COX-2 inhibitor) is recommended to slow disease progression in people with CF older than 6 years and with mild lung disease, also based on a 4-year trial that showed slower progression in lung disease [125]. However, the risk of side effects (mainly GI bleeding) has limited the use of this drug, even if benefits prevailed over the risks. Interestingly, in vitro and in vivo studies showed that ibuprofen may act as a CFTR corrector to restore CFTR trafficking in bronchial epithelial cells [126] and has anti-microbial actions against CF-related bacterial species [127,128]. Therefore, given this triple function, the risk–benefit ratio of ibuprofen administration to people with CF should be carefully evaluated. In contrast, selective COX-2 inhibitors are not recommended because of their cardiovascular side effects [129]. 

In addition to NSAIDs, other popular anti-inflammatory drugs such as steroids can interfere with AA metabolism. Indeed, steroids interfere with cPLA_2_, thus reducing both pro-inflammatory LT and PG production [130]. Despite that, their systemic long-term use in CF is discouraged due to the severe side effects. Furthermore, clinical results with aerosol steroids, used to minimize the harmful effects of systemic administration, were modest benefits [131]. 

Along these lines, drugs interfering with the LOX arm of the AA metabolism have been tested in CF to control inflammation. As mentioned, LT are elevated in CF and, in particular, LTB_4_ exerts potent pro-inflammatory actions by stimulating specific membrane receptors named BLT1 and BLT2 [132]. The potential therapeutic effects of the BLT1 antagonist have been tested in pediatric and adult patients with CF, but the trial has been interrupted due to serious pulmonary adverse events [133]. Another tested strategy in CF to reduce the excessive LTB_4_ aimed to reduce its production by inhibiting the LTA_4_ hydrolase (Figure 1). Acebilustat is a direct and selective LTA_4_ hydrolase inhibitor, that was found to be safe, well tolerated, and able to reduce inflammatory biomarkers (total leukocytes, PMN, total DNA and elastase in sputum) [134]. Despite these promising results in patients with CF treated with 15 days in phase I trial, in a 48-week phase II study, acebilustat failed to improve pulmonary function or to significantly reduce exacerbations, although a trend toward diminished exacerbations in people with CF at early stage of lung disease (FEV1 > 75) indicated a potential effect [135].

CysLT also activates two receptors, CysLT1 and CysLT2 [136]. Montelukast, a CysLT1 receptor antagonist, has shown some efficacy in reducing eosinophilic inflammation, improving clinical symptoms such as exercise tolerance, cough, and wheezing in people with CF, and lowering several inflammatory markers such as IL-8 and myeloperoxidase [137,138,139,140]. Similarly, Zafirlukast provided some benefits including improvement in chest radiograph appearance and physical examination, but not in respiratory function [141]. However, more powered studies are needed to evaluate their safety and efficacy.

Finally, as lipid peroxidation plays a crucial role in determining the distinctive lipid unbalance in CF, antioxidant administration could be proposed as a strategy to reverse the AA dysfunction. Indeed, administration of inhaled GSH to patients with CF significantly reduced PGE_2_ levels, along with increased the levels of CD4+ and CD8+ T cells. However, markers of oxidative stress such as 8-isoprostane were not changed [142], suggesting that the primary benefit of GSH inhalation in people with CF may be through modulation of the immune response rather than direct antioxidant effects.

A summary of the strategies tested to restore a balanced AA in CF in clinical trials is presented in Table 1.

## 3. Conclusions

Despite substantial advancements in care, chronic airway inflammation in CF persists as a vexing issue, characterized by a disproportionate, persistent, malfunctional activation of immune cells, mainly PMN, compared to the actual need for bacterial clearance. Concurrently, increasing evidence indicates an imbalance of AA metabolites in CF. Thus, there is an urgent need to investigate how alterations in the AA pathway could sustain chronic inflammation and how tailored therapeutic approaches could be developed to overcome the AA-dependent clinical complications in CF. 

The AA pathway, from its membrane-bound phospholipids to its metabolites, is a key player in immune regulation, as it orchestrates the different phases of the inflammatory response toward resolution in a tissue- and disease-specific manner [143], modulating the interactions between immune, resident, and endothelial cells. In line with this, a direct cause–effect relationship in CF has been established for the excessive airway concentration of LTB_4_ that induces further PMN adherence, superoxide production, degranulation, and neutrophil extracellular traps (NET) formation in the lung environment [72], thus sustaining PMN-mediated lung damage. Therefore, AA dysfunctions are a leading cause of CF pathology, suggesting that strategies to restore AA metabolism could be beneficial to hamper chronic inflammation and promote resolution. To this end, the integration of CFTR modulators, omega-3 fatty acids, enzyme inhibitors, receptor antagonists, and antioxidants holds some promise in preclinical studies for improving clinical outcomes through the restoration of physiological AA metabolism. However, failure of the BLT1 trial, likely due to the impairment of the beneficial physiological functions of LTB_4_, suggested that proper modulation of the AA pathway, despite inhibition of some enzymes or mediators, might be a better-tailored approach to preserve its physiology while dampening pathology. Importantly, inconsistent improvements in trials with DHA or acebilustat also suggested that drug timing, dosage, and target patients need to be carefully tailored to achieve durable drug efficacy. Thus, while some encouraging efficacies have been found in preclinical models, consistent evidence of improvement in patients remains elusive, highlighting the fact that future research is still warranted to restore a proper AA metabolism for people with CF.

## Figures and Tables

**Figure 1 pharmaceuticals-17-01185-f001:**
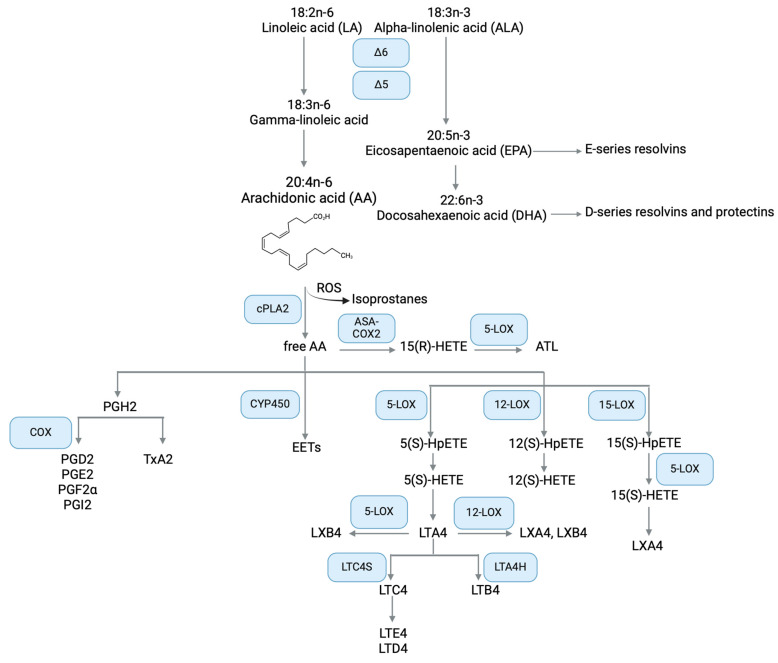
Simplified overview of the omega-6 and omega-3 PUFAs metabolism, focusing on the AA metabolic pathway components with relevance in CF. ALA metabolism to EPA, DHA, and SPM is also depicted. Created with BioRender.com.

**Figure 2 pharmaceuticals-17-01185-f002:**
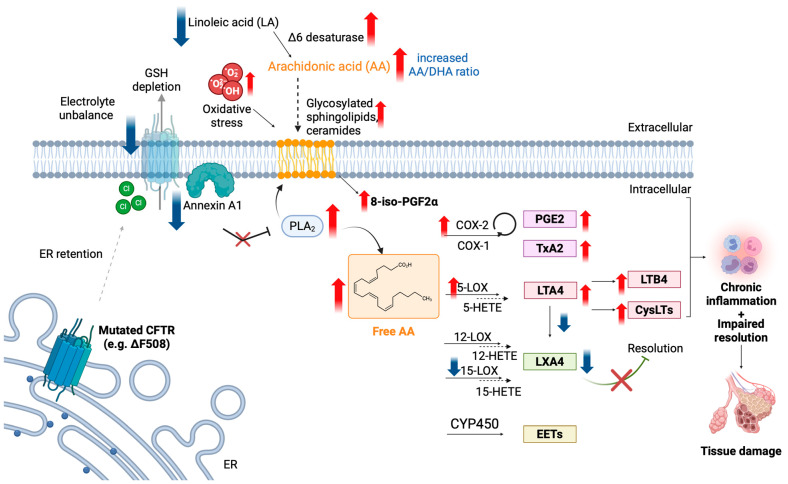
Dysregulation of the AA pathway in CF. Blue arrows indicate the enzyme/lipid mediator found downregulated in CF, while red arrows indicate the upregulation. See text for details. Created with BioRender.com.

**Figure 3 pharmaceuticals-17-01185-f003:**
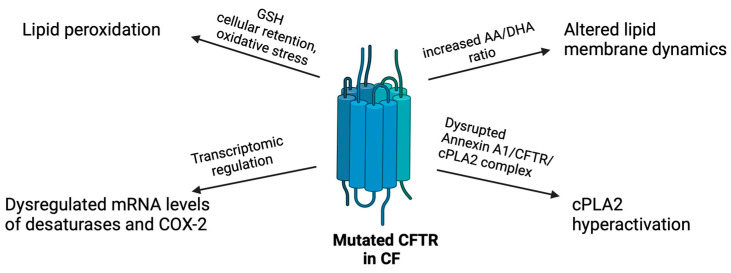
Infographic of the direct impact of mutated CFTR in the AA metabolism in CF. See text for details. Created with BioRender.com.

**Table 1 pharmaceuticals-17-01185-t001:** Therapeutic strategies proposed in CF clinical trials. For each therapeutic approach the cohort of patients with CF enrolled, the clinical outcome, and the respective reference are reported.

Strategy	Target	Clinical Outcome	Ref.
Ivacaftor	Patients with CF carrying at least one G551D CFTR mutation (*n* = 40).	AA and PGE2 levels decrease, but no effects on LA and DHA deficiency.	[117]
	CF pediatric patients (mean age = 11.7 years) (*n* = 11), for 12 months.	Reduced AA/EPA and AA/DHA ratios on red blood cell membranes.	[120]
DHA	Patients with CF (range 20–40 years) (*n* = 15), for 6 months.	Increased DHA and 17-OH DHA levels. Reduction in LTB4, 15-HETE and PGE2 levels, but no improvements of FEV1 and FVC.	[120]
	Five randomized controlled trials (*n* = 106).	Low DHA benefits and relatively low side effects.	[123]
	CF pediatric patients (Mean age = 11.7 years) (*n* = 22) for 12 months.	Improved FVC and FEV1, but PMN elastase, resolvin D1, IL-8, and TNF-alpha remained unaffected.	[124]
Ibuprofen	Patients with CF with mild lung disease (range 5–39 years) (*n* = 85), for 4 years.	Slowed lung disease progression, risk of GI bleeding.	[125]
Steroids (aerosol)	13 trials with patients with CF (range 6–55 years) (*n* = 506).	Modest general benefits.	[130]
BLT1 antagonist	CF pediatric patients (range 6–17 years) and adult patients (>18 years) with mild-to-moderate lung disease, for 24 weeks.	Serious pulmonary adverse effects.	[133]
Acebilustat	Adult patients with CF (range 28–55 years) (*n* = 17) with mild-to-moderate lung disease for 15 days.	Reduced sputum PMN and elastase. Trend toward reduction in serum C-reactive protein and sputum DNA.	[135]
	Adult patients with CF (range 18–35 years) (*n* = 199) for 48 weeks.	Absence of improvement in pulmonary function (FEV1) and pulmonary exacerbations. Trend toward reduced exacerbations in people with early lung disease.	[136]
Montelukast	Patients with CF (range 5–18 years) (*n* = 16), for 21 days.	Reduced eosinophilic inflammation, improved exercise tolerance, cough, and wheezing, decreased IL-8 and myeloperoxidase levels.	[137,138,139,140]
Zafirlukast	Patients with CF (range 20–32 years) (*n* = 25), for 4 months.	Benefits in chest radiograph appearance and physical examination without any improvement in respiratory functions.	[141]
Inhaled GSH	Patients with CF (range 18–39 years) (*n* = 17), for 14 days.	Reduced PGE2 levels, increased CD4+ and CD8+ T-cells levels. 8-isoprostanes levels remained unchanged.	[142]

## Data Availability

Not applicable.

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
