# Peer review of "Dysregulation of the Arachidonic Acid Pathway in Cystic Fibrosis: Implications for Chronic Inflammation and Disease Progression"

_pharmaceuticals, 2024, doi:10.3390/ph17091185_

Round 1
Reviewer 1 Report
Comments and Suggestions for Authors
This is an excellent review on an under reported topic of the arachidonic acid pathway in cystic fibrosis and how it feeds chronic inflammation and other CF phenotypes. The review is well-written, well-organized, and covers vital topics related to CF phenotypes and responses to modulators. The figures are clear and support the text nicely. Very well-done!
Author Response
Q1. This is an excellent review on an under reported topic of the arachidonic acid pathway in cystic fibrosis and how it feeds chronic inflammation and other CF phenotypes. The review is well-written, well-organized, and covers vital topics related to CF phenotypes and responses to modulators. The figures are clear and support the text nicely. Very well-done!
A1. We acknowledge the Reviewer for the very positive evaluation of our work.
Reviewer 2 Report
Comments and Suggestions for Authors
The current manuscript gives an in-depth overview of how the arachidonic acid pathway is dysregulated in cystic fibrosis. The authors effectively summarized the existing knowledge on this subject, highlighting both the biochemical and clinical consequences of this dysregulation. By revising the manuscript and addressing the suggestions for improvement, it can become even stronger and more effective at communicating the main findings and implications of the study.
- How does CFTR dysfunction affect arachidonic acid metabolism in cystic fibrosis?
- What are potential therapeutic interventions targeting arachidonic acid in CF?
- What are the implications of dysregulated arachidonic acid metabolism for chronic inflammation and disease progression in cystic fibrosis?
- It is recommended that the title be changed to: "Dysregulation of the Arachidonic Acid Pathway in Cystic Fibrosis: Implications for Chronic Inflammation and Disease Progression".
- The manuscript could benefit from more information about the data sources, sampling procedures, and analysis procedures used.
- The manuscript could benefit from more concise language throughout. This could include using shorter sentences and paragraphs, and avoiding unnecessary words and phrases.
- The manuscript could benefit from more tables and figures to help illustrate the main findings and implications of the study. This could include tables and figures that summarize the data, as well as diagrams and flowcharts that illustrate the research design and methodology.
- Some of the references could be more up-to-date and relevant to the specific research question and objectives.
Comments on the Quality of English LanguageThe manuscript could benefit from more concise language throughout. This could include using shorter sentences and paragraphs, and avoiding unnecessary words and phrases.
Author Response
Q1. How does CFTR dysfunction affect arachidonic acid metabolism in cystic fibrosis?
A1. We thank the reviewer for recognizing the importance of this important issue. Indeed, we have extensively discussed the published literature in Section 2.4, “Role of CFTR dysfunction in CF abnormal fatty acid metabolism”.
Q2. What are potential therapeutic interventions targeting arachidonic acid in CF?
A2. We agree with the Reviewer that is key to evidence tested therapeutic interventions in CF to address potential strategies to restore a balanced arachidonic acid pathway, which we reported in Section 2.5 “Therapeutic strategies to modulate AA metabolism in CF”.
Q3. What are the implications of dysregulated arachidonic acid metabolism for chronic inflammation and disease progression in cystic fibrosis?
A3. We thank the reviewer for raising this important point. Indeed, we have extensively discussed how arachidonic acid metabolism is affected in CF in Section 2 “Alteration of the AA pathway in CF, from precursors to metabolites”, which we have also summarized in Figure 2. However, as discussed in the Conclusion section, despite a large body of evidence showing alterations in arachidonic acid metabolism and the effect of arachidonic acid in regulating inflammation, a direct cause and effect relationship in CF inflammation has only been established for LTB4. Future studies are therefore warranted.
Q4. It is recommended that the title be changed to: "Dysregulation of the Arachidonic Acid Pathway in Cystic Fibrosis: Implications for Chronic Inflammation and Disease Progression".
A4. Suggested title perfectly matches the current review title.
Q5. The manuscript could benefit from more information about the data sources, sampling procedures, and analysis procedures used.
A5. We thank the reviewer for this suggestion. However, we trust that this kind of information could not improve the clarity of our review, which is based solely on the critical analysis of the reported literature on the topic found in PubMed.
Q6. The manuscript could benefit from more concise language throughout. This could include using shorter sentences and paragraphs, and avoiding unnecessary words and phrases.
A6. We agree with the Reviewer that more concise writing could improve the readability of the manuscript. We have revised the manuscript accordingly.
Q7. The manuscript could benefit from more tables and figures to help illustrate the main findings and implications of the study. This could include tables and figures that summarize the data, as well as diagrams and flowcharts that illustrate the research design and methodology.
A7. We agree with the reviewer that tables and figures are helpful to illustrate and summarize key points highlighted in the review. In fact, we have already included three figures and one table in the original version of the manuscript to help readers to quickly catch up with relevant findings. As described in Q5, we believe that the review does not need further explanation of the search criteria.
Q8. Some of the references could be more up-to-date and relevant to the specific research question and objectives.
A8. Thank you for this suggestion. We are aware that some of the references could be more up-to-date, but we have tried to include as much as possible the original publications that led to the described finding, despite referencing to new articles that indirectly mention to the original older work. We trust that readers will benefit more from this approach to quickly and directly recall the original publications discussed here.
Q9. The manuscript could benefit from more concise language throughout. This could include using shorter sentences and paragraphs, and avoiding unnecessary words and phrases.
A10. See point A6.
Reviewer 3 Report
Comments and Suggestions for Authors
This is a comprehensive review of the role of arachidonic acid (AA) metabolism in cystic fibrosis (CF), with a particular focus on the pathways related to AA metabolism and their alterations in the disease progression of CF. The discussion on potential therapeutic approaches targeting AA metabolism also provides a valuable contribution to understanding the current CF treatments. The manuscript is well-written and would be of great interest to a broad readership. I recommend it be published in Pharmaceuticals after addressing the following minor suggestions.
Line 204, the manuscript mentions "Higher levels were detected in patients with class I, II, and III mutations..." but does not specify what was higher. Please clarify what "levels" refers to.
Check typos, for instance, Line 318, “TRL2” should be “TLR2”.
The font size in Figures 1 and 2 is too small. Please increase it to improve readability.
Author Response
Q1. Line 204, the manuscript mentions “Higher levels were detected in patients with class I, II, and III mutations...” but does not specify what was higher. Please clarify what “levels” refers to.
A1. We acknowledge the Reviewer for the positive evaluation of our work and for raising this point. We referred to urinary prostaglandin metabolites, that we specified in the revised version of the manuscript.
Q2. Check typos, for instance, Line 318, “TRL2” should be “TLR2”.
A2. Thanks, we have amended this and other typos throughout the text.
Q3. The font size in Figures 1 and 2 is too small. Please increase it to improve readability.
A3. Thanks for the suggestion, we enlarged font size and corrected some typos in Figure 1.